# Sugarcane Rhizosphere Bacteria Community Migration Correlates with Growth Stages and Soil Nutrient

**DOI:** 10.3390/ijms231810303

**Published:** 2022-09-07

**Authors:** Zhaonian Yuan, Qiang Liu, Ziqin Pang, Nyumah Fallah, Yueming Liu, Chaohua Hu, Wenxiong Lin

**Affiliations:** 1Key Laboratory of Sugarcane Biology and Genetic Breeding, Ministry of Agriculture, Fujian Agriculture and Forestry University, Fuzhou 350002, China; 2College of Agricultural, Fujian Agriculture and Forestry University, Fuzhou 350002, China; 3Province and Ministry Co-Sponsored Collaborative Innovation Center of Sugar Industry, Nanning 530000, China; 4Fujian Provincial Key Laboratory of Agro-Ecological Processing and Safety Monitoring, College of Life Sciences, Fujian Agriculture and Forestry University, Fuzhou 350002, China; 5Key Laboratory of Crop Ecology and Molecular Physiology, Fujian Agriculture and Forestry University, Fuzhou 350002, China

**Keywords:** sugarcane, rhizosphere microbes, soil nutrients, growth stages, random forest, bacteria community

## Abstract

Plants and rhizosphere bacterial microbiota have intimate relationships. As neighbors of the plant root system, rhizosphere microorganisms have a crucial impact on plant growth and health. In this study, we sampled rhizosphere soil of sugarcane in May (seedling), July (tillering), September (elongation) and November (maturity), respectively. We employ 16S rRNA amplicon sequencing to investigate seasonal variations in rhizosphere bacteria community structure and abundance, as well as their association with soil edaphic factors. The results demonstrate that soil pH, total nitrogen (TN) and available nitrogen (AN) decrease substantially with time. Rhizosphere bacteria diversity (Shannon) and the total enriched OTUs are also significantly higher in July relative to other months. Bacteria OTUs and functional composition exhibit a strong and significant correlation with soil temperature (Tem), suggesting that Tem was the potential determinant controlling rhizosphere bacteria diversity, enriched OTUs as well as functional composition. Redundancy analysis (RDA) point toward soil total potassium (TK), pH, TN, Tem and AN as principal determinant altering shifting bacteria community structure. Variation partitioning analysis (VPA) analysis further validate that a substantial proportion of variation (70.79%) detected in the rhizosphere bacteria community structure was attributed to edaphic factors. Mfuzz analysis classified the bacterial genera into four distinct clusters, with cluster two exhibiting a distinct and dramatic increase in July, predominantly occupied by Allocatelliglobosispora. The stochastic forest model found the key characteristic bacterial populations that can distinguish the four key growth periods of sugarcane. It may help us to answer some pending questions about the interaction of rhizosphere microorganisms with plants in the future.

## 1. Introduction

Sugarcane is an important sugar crop that is widely planted all over the world [1,2]. It is widely used in many fields, such as food, energy, feed, etc. [3]. With the sharp increase in demand for sugar, more and more people are beginning to study how to increase sugarcane production in order to balance the easily growing demand [4,5,6].

The rhizosphere soil generally refers to the portion of soil found adjacent to the roots of living plants. The rhizosphere is subject to the influence of chemicals excreted by the roots of plants and the microbial community in this microzone [7]. Its domain varies for different plants and for stages and the morphology of roots. However, the rhizosphere microbial community means that close microbial associates in the root include those not touching roots but are heavily influenced by root exudates in the nearby soil [8]. They are closely related to the growth and development of plants [9,10,11]. Changes in the structure of the rhizosphere microbial community have an important impact on the circulation of substances and energy in the soil, the decomposition and synthesis of organic matter, etc. For instance, some scholars concluded that short-term N addition alleviates microbial nitrogen limitation and accelerates soil carbon (C) and N cycling [12], while some researchers also pointed out the following: the microbial community is not only the main biological agent of decomposition in soil but also a critical, albeit small, pool through which most of the organic matter in soil passes [13]. However, there are many factors affecting the change of rhizosphere microbial community structure. For example, the microbial community in the rhizosphere often differs across plants, as well as among genotypes within the same species [14,15,16]. Sometimes the structure and types of microbes in the rhizosphere of the same plant also vary greatly in different growth periods [17,18]. In addition, rhizosphere exudates provide nutrients and energy for the growth of rhizosphere microorganisms, which also has a certain influence on the abundance, type, distribution and growth of microorganisms [19,20] Soluble exudate mainly includes aliphatic and aromatic hydrocarbons, amino acids and sugars. Root exudates provide important sources of nutrients for the rhizosphere microbial community [21]. At the same time, plant growth is closely related to rhizosphere microbes. The physiological activities of rhizosphere microbes have obvious effects on soil properties, plant nutrient absorption and plant growth and development [22,23,24]. The population of rhizosphere microbes is also related to the health of plants and affects the yield and quality of crops to a large extent [25,26]. For instance, some rhizosphere microorganisms can produce hormones to promote the decomposition of soil organic matter, induce plants to increase resistance, etc., or indirectly promote plant growth by inhibiting the growth of pathogenic bacteria by competing for nutrients and occupying niches [27]. The dysregulation of the rhizosphere microbial community structure may lead to an increase in pathogenic bacteria, thereby reducing crop yields.

Currently, most of the studies on the interaction between sugarcane and rhizosphere microorganisms have focused on the symbiotic relationship between sugarcane roots and rhizobium and arbuscular mycorrhizal fungi. However, the relationship of sugarcane interactions with rhizosphere microorganisms during different growth stages remains unclear. Zhang et al. (2018), by a machine learning method, identified biomarker taxa and established a model of rice to correlate root microbiota with rice resident time in the field [17]. Therefore, understanding the changes in the rhizosphere bacterial community structure of sugarcane at different fertility stages is important to evaluate the influence of soil microbial diversity on the growth and development of sugarcane. Here, we aimed at providing a more comprehensive and extensive basis for modeling plant-root interactions with microorganisms. In this study, rhizosphere soil samples were collected from sugarcane in the seeding, tillering, elongation and maturity stages. High-throughput sequencing technology was applied to 16S rRNA of soil samples using the Illumina sequencing platform to explore the structural diversity and changes of rhizosphere soil bacterial communities in four key stages of sugarcane and to discuss the variation of rhizosphere microbiota during sugarcane cultivation. Our work is important for evaluating the influence of soil microbial diversity on sugarcane growth and development.

## 2. Results

### 2.1. Soil Physiochemical Properties

Rhizosphere soil physicochemical properties were significantly different (*p* < 0.05) among the samples collected in May, July, September and November during sugarcane seedlings, tillering, elongation and maturity stages, respectively (Figure 1, Appendix A). Soil pH, TN and AN temporal variations were seasonal dependents (Figure 1A,D,G). Moreover, soil OM and TP showed significant improvement in the warmer months, especially in September, while soil Tem was profoundly enhanced (*p* < 0.05) in both July and September (Figure 1B,C,E). Compared to November, soil AK considerably peaked (*p* < 0.05) in May and July; however, soil AP fluctuated throughout the entire season (Figure 1F,G,I).

### 2.2. Rhizosphere Bacteria Community Structure, Diversity and Richness

The samples were collected in May, July, September and November during sugarcane seedling, tillering, elongation and maturity stages, respectively, and were used to investigate bacteria community structure (Figure 2A). Unconstrained principal coordinate analysis (PCoA) of Bray–Curtis distance and Anosim analyses further demonstrated that the dissimilarity of rhizosphere bacterial community structures varied considerably from one month to another (Figure 2B and Table 1). It was also observed that the species richness and diversity of bacteria community structure were more diverse throughout the different time points (Appendix A). The estimation of bacteria diversity (Shannon) was more pronounced in the warmer season (July), followed by September and November compared to May (Figure 2C). Bacteria phyla relative abundance showed that Proteobacteria (24.85–29.28%), Actinobacteria (15.53–19.09%), Chloroflexi (7.63–18.89%), Acidobacteria (6.53–11.05%), Firmicutes (5.59–37.36%) were the dominant bacteria, followed Bacteroidetes (1.51–2.69%), Gemmatimonadetes (1.50–6.95%), Planctomycetes (0.99–2.60%) and Saccharibacteria (0.65–3.01%) (Figure 2D). Meanwhile, source-tracking analysis was adopted to identify the potential sources of bacteria relative abundance from one period of the sugarcane growth stage to another using the Source Model of Plant Microbiome (SMPM) tool. The result revealed that the highest number of bacteria was detected from September to November (97.76%), followed by May to July (88.48%) and July to September (87%) (Appendix A). A Venn diagram analysis was then employed to unveil the unique and overlapped bacteria OTUs detected in the different samples. The analysis revealed that in May, July, September and November, 92, 92, 54 and 116 unique bacteria OTUs were identified, respectively. Despite the low number of enriched bacteria OTUs that were unique in July, the total OTUs (7392) detected during the warmest season (July) were considerably more, followed by 6957, 6841 and 6673 OTUs in September, November and May, respectively (Figure 2E).

Regression analysis was adopted to assess the relationship between soil physiochemical properties and PCoA axes 1 and 2 scores as a proxy for bacterial community structure, Shannon and ACE indices (Figure 3a–l) based on the significance of the interactions between soil edaphic factors and different bacteria indicators displayed on the heat map (Figure 3m). The analysis demonstrated that soil TN (r = −0.57), AK (r = −032), AN (r = −0.43) and TK (r = −0.69) were negatively associated with Shannon (*p* < 0.05) (Figure 3a–d). Moreover, soil pH (r = −0.81), TN (r = −0.63), TK (r = −0.63) and AN(r = −0.68) had a negative correlation with PC1 (Figure 3i–l). On the other hand, soil TN (r = 0.39), AN (r = 0.34), TK (r = 0.43) and pH (r = 0.63) revealed a positive association with PC2 and ACE, respectively (Figure 3e–h).

### 2.3. Correlations of Soil Properties with Bacterial Communities

Redundancy analysis and linear regression revealed that variation in bacterial community composition was significantly correlated with specific soil properties (Figure 4A,D and Appendix A). *Actinobacteria* and *Chloroflexi* were negatively correlated with soil TK, TN content and positively correlated with soil TP content, while *Acidobacteria*, *Nitrospire* and *Cyanobacteria* tended to prefer high soil temperature and low AK content soil conditions. Additionally, as revealed by the RDA biplot and the further permutations test, soil pH, TK and TN content were the key predictors of bacterial community structure. Indeed, soil temperature and AN content also played significant roles. Meanwhile, VPA was then performed to quantify the proportion of variance in the bacteria community structure induced by soil edaphic factors. The analysis demonstrated that a substantial proportion (70.79%) of the variance observed in rhizosphere bacteria community structure was attributed to soil edaphic factors (Appendix A). Mantal’s test verified that temperature plays an important role in the composition and function of sugarcane rhizosphere bacteria in different seasons (Figure 4C). In addition, the RDA of bacterial function also showed that May was distinct from the other seasons based on the different directions of change in the TN, TK, AN and pH. It also showed that the function of bacteria differs not only in different growth stages but also by the physicochemical properties of the soil (Figure 4B). In the tillering stage, there was less metabolism and more environmental information processing. In addition, the cellular processes were positively correlated with pH, AN and TK but negatively correlated with OM; metabolism was also more inclined to high TP content and low TN, AK concentration.

### 2.4. Soil Environmental Variables and Rhizosphere Bacteria Co-Occurrence Network

Co-occurrence networks were constructed using bacteria OTUs obtained from the various seasons. The network topology parameters calculated by Gephi software (Appendix A) showed that there were significant differences in the number of network nodes, the number of connections, aggregation coefficient, characteristic path length, network density and the average connectivity during the different seasons (Figure 5A–D). The relationship between bacterial community composition and their relationship with soil physiochemical properties was also assessed (Figure 5E). The result showed that the network in May consisted of 1099 nodes by 5922 edges, with the highest positive correlation of 83.06% and the lowest negative correlations of 16.94% compared to July, September and November. In July, the network consisted of 1705 nodes by 29,822 edges, with the second highest positive association of 70.99% and 29.01% negative interaction. The analysis also showed that the total nodes and edges in September were 1276 and 16,005, respectively, with a positive interaction of 71.91% and a negative interaction of 28.09%. While in November, the total nodes and edges during the sugarcane maturity stage were 1128 and 62,237, respectively, with 57.64% and 42.36% positive and negative correlation, respectively (Appendix A). In addition, the network analysis showed that a substantial portion of key bacteria genera displayed an extensive negative correlation with soil edaphic factors (Figure 5E).

### 2.5. Differential Rhizosphere Bacteria Abundance Detected during Various Seasons

Rhizosphere bacterial community structures were apparently changed and varied somewhat during the course of the year as environmental factors and sugarcane growth stages changed, but the dominant taxa remained Acidobacteria, Actinobacteria, Chloroflexi, Firmicutes and Proteobacteria. We further performed LEfSe analyses for 84 samples to examine which taxa differed most in different stages of sugarcane. The LEfSe cladogram and LDA (Figure 6A and Appendix A) showed that many taxa were common between the different stages of sugarcane (yellow circle), but there were still some specific differences. For example, the relative abundance of Firmicutes in the rhizosphere of sugarcane was higher than in other seasons, while the relative abundance of Acidobacteria was significantly higher in July and Sep. At the same time, through LEfSe, the important biomakers of each growth period were also visualized. They were Micrococcaceae, Massilia and Oxalobacteraceae in May; *Nitrospira*, *Varlibacter* and Deltaproteobacteria in July. Furthermore, Acidobacteriaceae and Rhodocyclaceae were in September, and Kineosporiaceae, Actinobacteria, Nitrolancea, Sphaerobacterales, Sandaracinus, Mizugakiibacter and Opitutaceae were in November (Figure 6A). Regarding KEGG, 11 pathways were significantly different in second-level pathways (LDA > 2.5, *p* < 0.05, Figure 6B), including 6 pathways with significant differences in May, such as “Two-component system”, “Signal transduction”, “Nitrogen metabolism”, “Dioxin degradation” and other pathways, 2 pathways in July (Endocrine system and Nitrotoluene degradation), 2 pathways in Sep (sphingolipid metabolism and pentose and glucuronate interconversions) and 1 pathway (biosynthesis of unsaturated fatty acids) in Nov, respectively. At the first level, most of the secondary differential metabolic pathways focus on metabolism.

Mfuzz clustering analysis of the relative abundance of microorganisms’ time-course data identified four clusters with significant change tendencies (Figure 7). Cluster 2 presented an increasing trend (Figure 7B). Specifically, the relative abundance of bacteria exhibited an increase from May to Sep; subsequently, the relative abundance increased significantly from Sep to Nov. In addition, the change tendency of relative abundance of bacteria in Cluster 2 (Figure 7B) at different time points was evidently opposite to those observed in Cluster 4 (Figure 7D). The relative abundance of bacteria in Cluster 4 decreased significantly from May to July; subsequently, the levels decreased slightly from July to Nov. It is of note that the change tendency of relative abundance in Cluster 1 (Figure 7A) at different time points is analogous to that observed in Cluster 3 (Figure 7C), but the timing of the turning point is different. Additionally, differential bacteria genera were screened. A total of 605 genera were identified, including 141 different genera of bacteria in Cluster 1164 different genera in Cluster 2201 different genera in Cluster 3 and 99 different genera in Cluster 4. In addition, Mfuzz analysis based on all bacterial genera in the sugarcane rhizosphere showed significant differences in the variation of bacterial genera from season to season (Appendix A).

The list of the top 26 bacterial taxa at the class level in the rhizosphere soil of sugarcane, in order of time-discriminatory importance and abundance, is shown in Figure 8A,B. They distinguished the specific characteristics of bacterial populations in the rhizosphere soil of sugarcane at different stages. Most biomarker taxa showed a high relative abundance at the tillering and ripening stages of sugarcane. For instance, TM6-Dependentiae, Thermomicrobia, Norank-FCPU426, and Gemmatimonadetes all started to accumulate in sugarcane rhizosphere soil during seeding and elongation stages and kept high levels at the sugarcane maturity stage. In addition, bacteria norank-Latescibacteria, lgnavibacteria, and Dehalococccoidia, which have an important discrimination degree, accumulated to the highest relative abundance during the tillering stage.

## 3. Discussion

In recent years, plants rhizosphere microbiota have received more attention due to the rapid development of next-generation sequencing, and many investigations have confirmed that rhizosphere microbiota are involved in many key processes of plant growth, including boosting plant immunity [28], pathogen abundance of roots [29], plants’ nutrient acquisition [30] and stress tolerance [31]. Generally, rhizosphere microflora is considered to be an important factor induced by environmental and genetic factors. Microorganisms are implicated in plant growth and health [32,33].

A growing amount of evidence has shown that soil microbiota can influence the health of plants and soil. For instance, Yuan et al. (2020) used high-throughput DNA sequencing and machine learning based on big data to compare the differences in the microbiota of diseased versus healthy soil, and the potential relationship between soil microbiota and plant health was also demonstrated [34]. Additionally, mounting evidence has suggested that various environmental factors and soil management practices such as agrochemical input can have a consequential effect on soil physicochemical indicators, which in turn could affect soil microbial community diversity. In a recent study, Pang et al. (2022), documented that N-only treatment significantly reduced soil NO_3_--N and nitrification potential, which in turn had a considerable impact on some potential nitrogen-fixing bacteria community diversity, namely, Mesorhizobium and Rhizobacter [35]. According to our results, the physicochemical indicators of sugarcane rhizosphere soil showed significant variability in different periods (Figure 1, Appendix A). One point worth noting was that soil pH gradually decreased over time, which may be related to the accumulation of rhizosphere microbial activity and certain root secretions in sugarcane [36,37]. The highest relative abundance of five major phyla, Acidobacteria, Actinobacteria, Chloroflexi, Proteobacteria and Firmicutes, was found to be the dominant flora in the rhizosphere soil of sugarcane during growth. In the study by Liu et al. (2018), the highest abundance of Proteobacteria was found in soybean rhizosphere soil, followed by Acidobacteria and Actinomycetes [38]. In a related study, the microbial community diversity from the roots of G. conopsea and related soil samples from different biogeographical regions at two distinct altitudes were explored at the reproductive and vegetative growth stages. It was observed that developmental stage, compartment and geographical location were factors influencing microbiome community diversity in G. conopsea [39]. At the same time, the reasons for the changes in the abundance of sugarcane inter-rhizosphere flora at different fertility periods, we speculated that it may be due to the specificity of the crop growth period, environmental factors and the variation of root secretions at different fertility periods [40,41]. In addition, the apparent separation of the first axis in PCoA and the model of source analysis (Appendix A) within different stages reinforced the variability and transmission of sugarcane rhizosphere flora between seasons. According to our correlation heat map data, pH, TN and TK were strongly correlated with the Alpha diversity index of bacteria and were important drivers of the rhizosphere bacterial community in sugarcane, which was consistent with the findings of Tayyab et al. (2021) [42]. Meanwhile, the strong negative correlation between pH and PC1 may suggest a potential interaction between sugarcane fertility, pH and the rhizosphere microbial community, so RDA and Mantel’s test were used to further investigate the correlation between rhizosphere bacterial compartment and environmental factors. It was found that pH, TN, Tem, AN and TK were strongly correlated in RDA not only with the major rhizosphere bacterial communities of sugarcane but also with the predicted functions of rhizosphere microorganisms.

Mantel’s test further confirmed the existence of this correlation while showing the strong effect of soil temperature variation on the rhizosphere microbial community in different seasons. This is analogous to previous studies of soil fungi as follows: Soil abiotic factors were found to be the major determinants of soil fungal community composition, such as Zhang et al. (2016), who reported that soil pH, organic C, organic C and organic N were significantly related to soil fungal community composition in Arctic deserts [43]. Furthermore, Tedersoo et al. (2014) concluded that soil pH was a critical predictor of soil fungal community composition in 365 soil samples sampled from six continents [44]. This is coincidentally consistent with the strong correlation between pH and bacterial flora in our results. In general, we found the soil factors pH, Tem, TN, AN and TK played an important role in driving the variability of rhizosphere bacterial communities and were important predictors during different sugarcane fertility periods. The diversity and composition of rhizosphere bacterial communities were strongly related to the sugarcane growth stages [45]. Owing to the influence of seasonal changes, environmental factors affected the amount and composition of root exudates, which caused changes in the rhizosphere soil micro-environment [46]. We found that obviously different modules containing a high proportion of OTUs in the co-occurrence networks can reflect different stages of sugarcane growth. Co-occurrence networks not only showed differences in modules among complex microbial communities(Figure 5A–D) but also showed that critical bacteria were highly relevant to the soil physicochemical properties (Figure 5E), which can partly assess and indirectly determine ways in which critical bacteria affect sugarcane growth [47].

LefSe analysis was used to find rhizosphere bacterial biomarkers in sugarcane during different fertility periods to better distinguish the variability between seasons. In total, 37 biomarkers were identified in LEfSe analysis, including 4 in May, mainly from Massilia, Oxalobacteraceae and Micrococcaceae, where it was noted that Massilia colonized and proliferated on the seed coat, radicle and roots. High variation in Massilia abundance was found in relation to plant developmental stage, along with sensitivity to plant growth medium modification (amendment with organic matter) [48]. At the same time, Massilia peaked (up to 85%) at the early stages of the succession of the root microbiome. This may be an important reason for Massilia to be a biomarker of sugarcane seedlings. In addition, *Micrococcaceae* is a genus of bacteria widely found in nature [49]. There were 13 biomarkers in July, mainly SBR2076, *Nitrospira*, *Variibacter* and Deltaproteobacteria, but only 2 biomarkers in September, Rhodocyclaceae and Acidobacteriaceae. Nitrospira is a chemolithoautotrophic nitrite-oxidizing bacterium. Nitrospira-like bacteria take up inorganic carbon (such as HCO_3_^−^ and CO_2_) as well as pyruvate under aerobic conditions [50]. Furthermore, Lueders et al. (2004) showed that environment-dependent metabolic versatility and the presence of nitrogen-fixing bacteria are affiliated with a range of taxa, encompassing members of many bacterial genera, including Deltaproteobacteria [51,52]. However, Rhodocyclaceae is an important group of bacteria involved in the degradation of ethylbenzene and Acidobacteriaceae play an important role in the formation of acetic acid and the utilization of sugar [53,54]. We speculated that this is importantly linked to certain key physiological activities in the roots of sugarcane during the elongation period. Meanwhile, the largest number of biomarkers in November was 18, mainly in Actinobacteria, Kineosporiaceae, Kineosporiales, Nitrolancea, Sphaerobacteraceae, Sandaracinus, Mizugakiibacter and Opitutaceae. The diversity of biomarkers implied the complexity and variability of physiological processes in sugarcane at different stages. This is similar to the results of the rhizosphere bacteria population with characteristics and discrimination in the stochastic forest model.

In addition, we used Mfuzz analysis to observe the dynamic variability of rhizosphere differential bacterial changes in sugarcane over time. Mfuzz uses a soft clustering algorithm, which reduces the interference of noise on clustering results to some extent compared with other algorithms such as K-means [55]. Mfuzz analysis clustered sugarcane rhizosphere differential bacterial genera into four categories. *Actinoplanes*, *Asticcacaulis,* etc. with significant changes in relative abundance in Cluster 1(Figure 7A), *Nitrosomonas*, *Paenibacillaceae* and *Clostridiales* were the hub bacterial genera (Figure 7B, red) in Cluster 2. These bacterial genera are closely related to the ammonia-oxidizing, biofilm formation and utilization of organic carbon of sugarcane at the tillering stage [56,57,58]. Cluster 3: Allocatelliglobosispora and so on (Figure 7C); Cluster 4: *Tahibacter*, *Bosea*, *Sinomonas,* etc. (Figure 7D). There was a clear regularity in the trends developed by the differential bacterial genera over time, and this regular variability further implies an intrinsic link between sugarcane growth and development and rhizosphere microbial activity at different times. In summary, this study established a sugar cane growth model and provided the reference for an in-depth understanding of how changes in the rhizosphere bacterial communities respond to different growth stages of plants.

## 4. Materials and Methods

### 4.1. Sample Collection and Soil Physiochemical Analysis

The soil samples used were collected from the Baisha Experimental Station of the National Sugarcane Research Center of Fujian Agriculture and Forestry University (119°06′ E, 26°23′ N, subtropical monsoon climate with an average annual temperature of 19.5 °C, and an average annual precipitation of 1673.9 mm). Samples were collected from seedling (accessed on 4 May 2017), tillering (accessed on 2 July 2017), elongation (accessed on 17 September 2017) and maturity (accessed on 29 November 2017) stages. A total of 84 samples (4 periods, 21 replicates) were collected from 21 sampling points in sugarcane fields using the “S-shaped sampling method” [59], and the soil adhering to plant roots was removed with a small sterile brush according to the shaking method of Riley and Barber [60], and then mixed and homogenized by passing through a <2 mm sieve to remove visible residue and stones. The fresh soil sample was portioned into two subsamples and stored in polyethylene bags. A portion was stored at 4 °C to measure the physiochemical properties of soil. The remaining portion was then stored at −20 °C for DNA extraction and microbial analysis. Soil suspension with water (1:2.5 WV^−1^) was prepared in order to measure soil pH by adopting a pH meter (PHS-3C, INESA Scientific Instrument Co., Ltd., Shanghai, China) [61]. The soil temperature (Tem) was calculated using soil temperature detector (Model: JC-TW, China). Soil total nitrogen (TN) and organic matter (OM) were estimated as described by Kjeldahl digestion and determined by oil bath–K_2_CrO_7_ titration approach [62,63]. C:N is the ratio of soil total nitrogen to organic matter. Soil total potassium (TK) and total phosphorus (TP) were measured using digestion HF-HClO_4,_ followed by flame photometry and molybdenum-blue colorimetry, respectively [64,65]. Available potassium (AK) was extracted using ammonium acetate and measured by flame photometry [66]. Available phosphorus (AP) was estimated using sodium bicarbonate and measured by employing the molybdenum blue method [67].

### 4.2. Extraction and Electrophoretic Detection of Soil DNA

Soil DNA extraction was carried out using a Power Soil DNA Isolation Kit (MoBio Laboratories Inc., Carlsbad, CA, USA) following the manufacturer’s instructions. Agarose gel electrophoresis was employed for qualitative detection, and a NanoDrop 2000 spectrophotometer (Thermo Scientific, Waltham, MA, United States) was adopted to measure the DNA concentrations and later stored in a −20 ℃.

### 4.3. 16S rRNA PCR Amplification and Sequencing

To examine the rhizosphere bacterial microbiota. Bacterial communities were characterized by pyrosequencing 16S rRNA gene amplicons derived from the PCR primers 338F/806R [8,68] and sequenced using the Illumina MiSeq platform. PCR conditions: 95 °C for 3 min, followed by 35 cycles of 95 °C for 30 s, 55 °C for 30 s and 72 °C for 45 s, with a final extension at 72 °C for 10 min (GeneAmp 9700, ABI, California CA, USA). PCR reactions were carried out in triplicate in a 20 μL mixture containing 2 μL of 2.5 mM deoxyribonucleoside triphosphate (dNTPs), 4 μL of 5× Fast Pfu buffer, 0.4 μL of Fast Pfu polymerase, 0.4 μL of each primer (5 μM) and template DNA (10 ng). QuantiFluor™-ST (Promega, Madison, WI, USA) was used for quantification.

### 4.4. Data Quality Control and Filtering

The extracted data were saved in fastq format to distinguish the data of each sample according to the index sequence, and the paired reads were spliced (merged) into one sequence according to the overlapping relationship between PE reads, while the quality of the reads and the effect of merge were filtered for quality control, and the valid sequences were obtained according to the barcode and primers at the first and last ends of the sequences [69]. The pair-end double-end sequence splicing (Flash, 1.2.11). Generate an abundance table for each taxonomy (QIIME, 1.9.1) [70]. The raw sequencing data have been deposited in the public database NCBI, registration number: PRJNA721464.

### 4.5. Statistical Analysis

Based on the OTU results, the rarefaction curve and Shannon index curve (Appendix A) were analyzed using Mothur [71]. Statistical analysis was conducted using DPS and SPSS 13.0 [72,73]. Using Adobe Illustrator CS6 (AI) to complete the construction of sugarcane growth model [74]. The Venn diagram [75], box plots [76] and correlation heatmap [77] as well as variation partitioning analyses (VPA) were completed using the Vegan package of R (Version 3.6.0). To further quantify the difference in bacteria community structure, nonparametric statistics based on the Bray–Curtis dissimilarity index were performed using the rhizosphere bacteria OTU. An principal coordinate analysis (PCoA) and an analysis of similarities (Anosim) were conducted to test if there was indeed a significant difference in rhizosphere bacteria community structure among different growth stages of sugarcane [78]. At the same time, to visually interpret community dissimilarity and investigate the relationship between bacteria community data and physicochemical data, redundancy analysis (RDA) was selected, and the significance of total physicochemical factors was tested with Monte Carlo permutations (permu = 999) [79]. Mantel’s test for testing the correlation between environmental factors and different matrices [80]. A co-occurrence network was constructed to expound the interaction between the genera of bacteria community structure as well as their relationship with soil edaphic factors. The network was visualized by conducting a correlation matrix and calculating all potential pairwise Spearman’s ranks by employing Cytoscape version 3.2.0 [81]. LEfSe was adopted to elucidate the biomarkers in each group. Kruskal–Wallis sum-rank test was applied to detect features with significant differential abundance, and Wilcoxon rank-sum test was subsequently used to investigate consistency of bacteria characteristics using a set of pairwise tests among sub-classes. Finally, linear discriminant analysis (LDA) was employed to estimate the effect size of each differentially abundant feature using the default effect size threshold of 2 (log10) (LDA score) [82]. Source-tracking analysis was employed to evaluate the potential sources of bacteria community from one sugarcane growth stage to another using Source Model of Plant Microbiome (SMPM). Both R software (http://www.r-project.org/, accessed on 16 March 2022 ) and Bioconductor (http://www.bioconductor.org/, accessed on 20 March 2022 ) package “Mfuzz” were employed to examine the expression cluster time-series data of bacteria genera based on fuzzy c-means. The fuzzification parameter was adjusted to m = 2 and the number of clusters to c = 4 to maintain the soft clustering of the Top-1000 list [55]. In order to obtain the discriminative performance of rhizosphere bacterial taxa at different growth stages of sugarcane, we used the default parameter of R realization of the algorithm (R package ‘random forest’, ntree = 1000) to regress the relationship between the relative abundance of bacterial taxa at the taxonomic level and the seasonal development of sugarcane.

## 5. Conclusions

The rhizosphere microbial community is an important player in the plant-soil ecosystem. In our present field experiment to establish a rhizosphere model of sugarcane, the highest relative abundance of five major phyla, Acidobacteria, Actinobacteria, Chloroflexi, Proteobacteria and Firmicutes, was found to be the dominant flora in the rhizosphere soil of sugarcane during growth. Soil factors pH, Tem, TN, AN and TK played an important role in driving the variability of rhizosphere bacterial communities and were important predictors during different sugarcane fertility periods. Meanwhile, co-occurrence networks showed differences in modules among complex microbial communities; 37 biomarkers were identified in the LEfSe analysis. These biomarkers are likely to be involved in or influence physiological processes during specific reproductive stages of sugarcane. In addition, Mfuzz analysis clustered the sugarcane rhizosphere differential bacterial genera into four categories and demonstrated the dynamic changes over time. Overall, this study may help us to answer pending questions about the interaction of rhizosphere microorganisms with plants in the future.

## Figures and Tables

**Figure 1 ijms-23-10303-f001:**
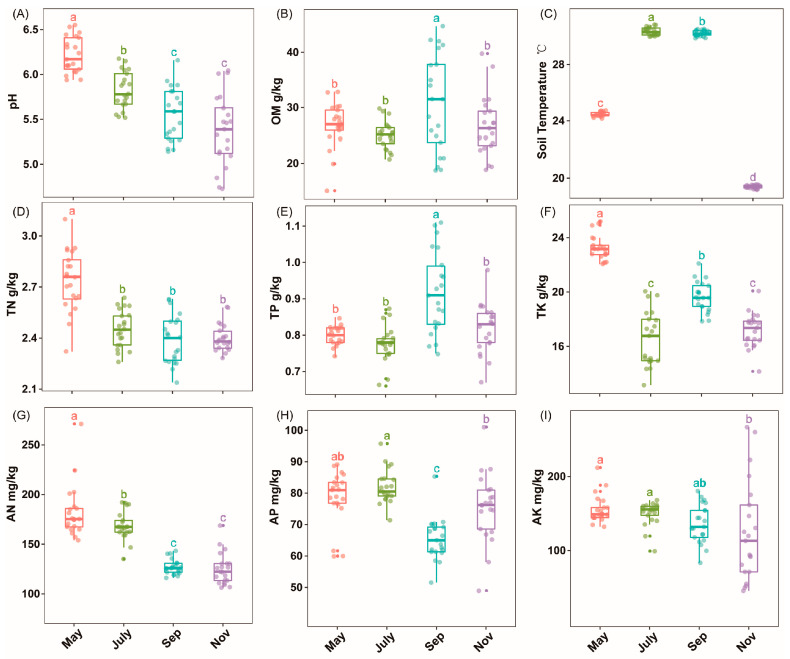
The box plot shows soil physiochemical property, namely, pH (**A**), OM, soil organic matter (**B**); TN, total nitrogen (**D**); TP, total phosphorus (**E**); TK, total potassium (**F**); AK, available potassium (**G**); AP, available phosphorus (**H**); AN, available nitrogen (**I**); Tem, soil temperature collected in May, July, September and November during sugarcane seedlings, tillering, elongation and maturity stages, respectively (**C**). Different letters indicate significantly different among the different time points (*p* <  0.05, ANOVA, Tukey HSD).

**Figure 2 ijms-23-10303-f002:**
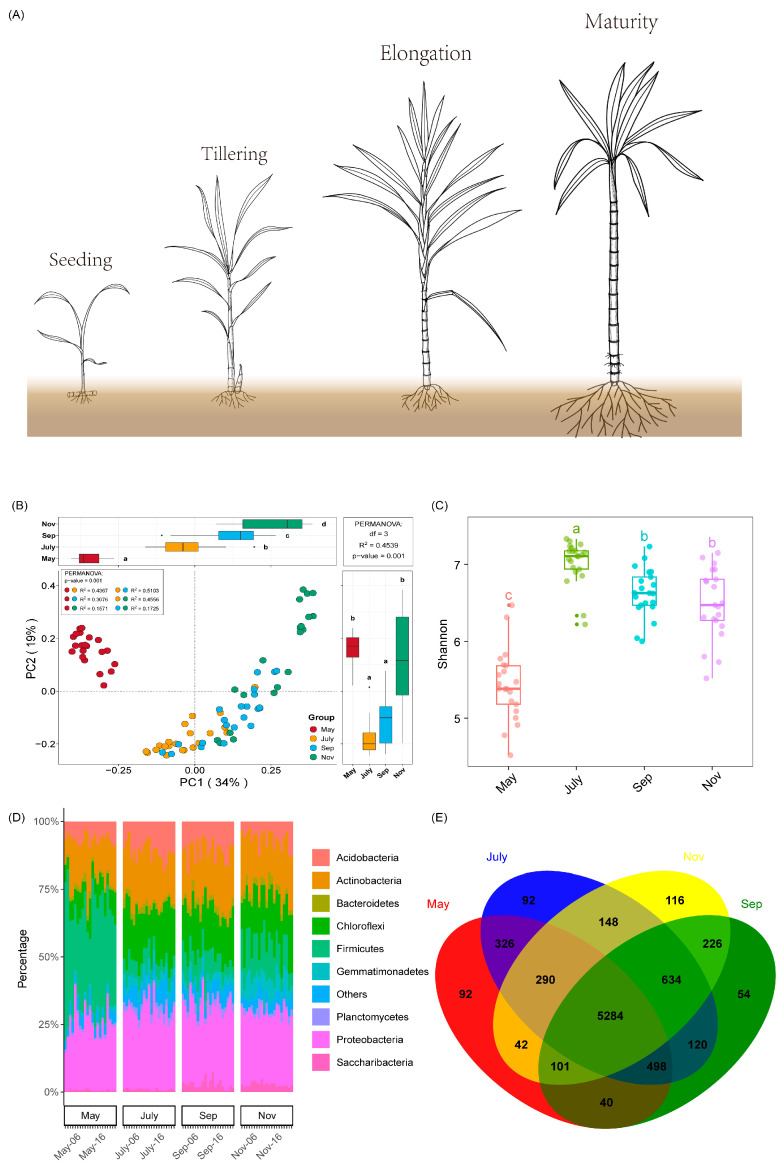
(**A**) Seasonal growth model of sugarcane. (**B**) Unconstrained PCoA (for principal coordinates PC1 and PC2) with Bray-Curtis distance. (**C**) The Shannon index of soil microbial populations in roots of sugarcane at different growth stages. (**D**) Phylum-level distribution of rhizosphere soil in different growth stages of sugarcane. (**E**) Venn diagram of the core OTU of the rhizosphere soil of sugarcane at different growth stages. Different letters indicate significantly different among the different time points.

**Figure 3 ijms-23-10303-f003:**
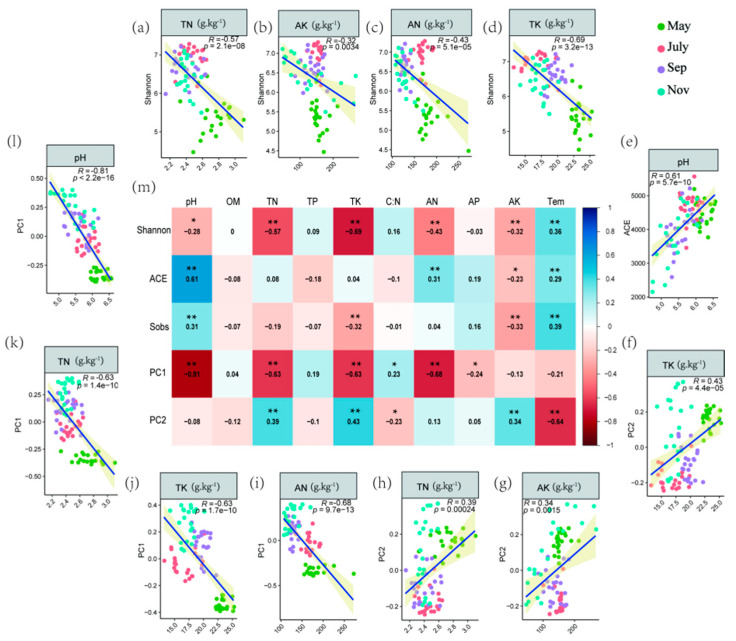
(**a**–**l**) Linear relationships among selected soil microbial properties and physicochemical variables. Their bivariate correlations were still significant after FDR adjustment (*p* < 0.05). (**m**) Bivariate correlations among important variables and soil physicochemical. The color indicates the direction of correlation, and the number represents the correlation coefficient, * *p* < 0.05, ** *p* < 0.01.

**Figure 4 ijms-23-10303-f004:**
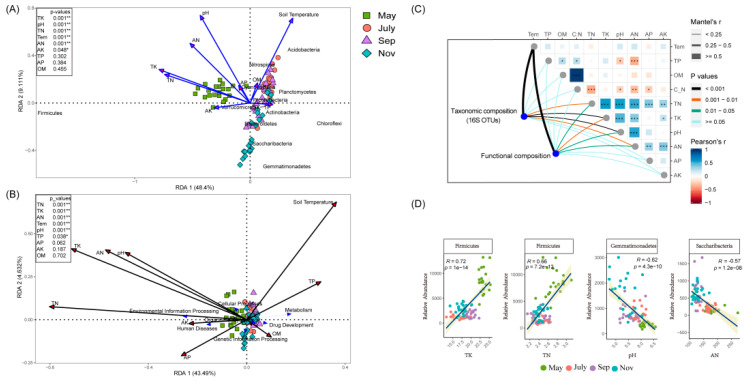
RDAs analysis between soil edaphic factors and bacteria phyla (**A**) and KEGG-L1 (**B**), sampled in May, July, September and November. In the top-left, the soil soil edphic factors were fitted to the ordination plots using a 999 permutations test (*p*-values). (**C**) Depicts pairwise comparisons of soil edaphic factors displayed with a color gradient. Taxonomic composition (16S OTUs) and functional composition correlation with soil edaphic factors using Mantel-tests. Each edge width matches with the Mantel’s statistic for the corresponding distance correlations. (**D**) Regression analysis of important bacteria and key soil factors at the phylum level. * *p* < 0.05, ** *p* < 0.01, *** *p* < 0.001.

**Figure 5 ijms-23-10303-f005:**
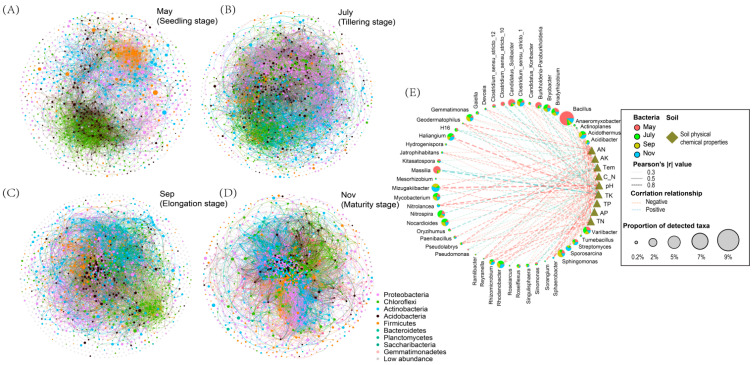
Co-occurrence networks analysis between rhizosphere bacteria OTUs and the different growing seasons of sugarcane. ((**A**) May, seedling stage; (**B**) July, tillering stage; (**C**) September, elongation stage and (**D**) November, maturity stage), and the rhizosphere bacteria network have different structural characteristics at four growth stages. (**E**). Co-occurrence network depicting relationship between rhizosphere bacteria and soil physiochemical properties.

**Figure 6 ijms-23-10303-f006:**
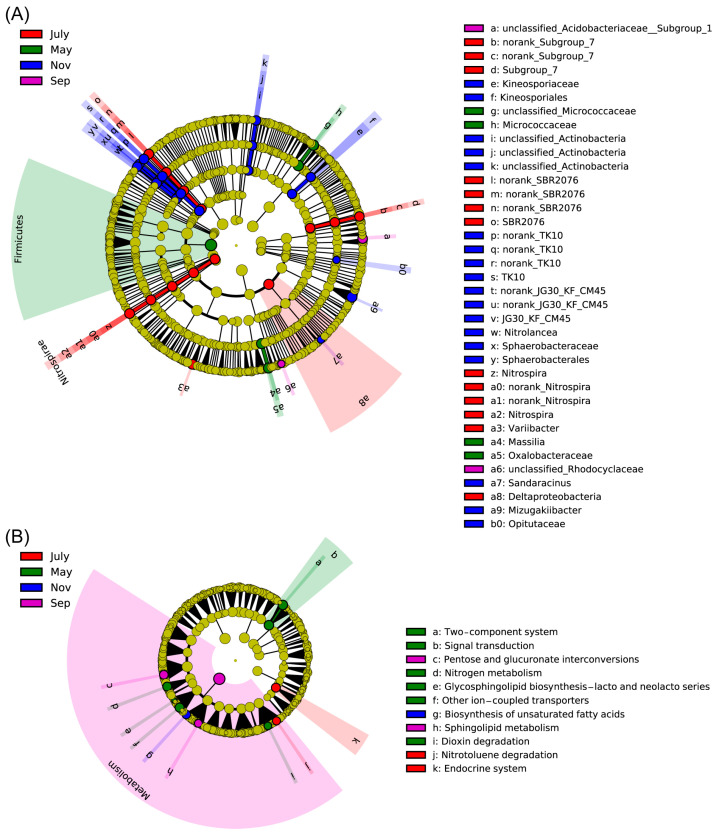
Cladogram plotted from LEfSe comparison analysis indicating the taxonomic representation of statistically and biologically consistent differences of identified biomarkers among different time point. Differences were represented in the color of the most abundant taxa (green stands for May, red for July, purple for Sep, blue represents Nov and yellow were non-significant). Each circle’s diameter was proportional to the given taxon’s relative abundance. (**A**) Cladogram indicating the phylogenetic distribution of bacteria lineages under seasonal changes of sugarcane; lineages with LDA values higher than 3.5 were displayed. Circles represent phylogenetic levels from phylum to genus from the inside outwards. (**B**) KEGG level 1~3 functional pathways differentially abundant by different stages. Differentially abundant KEGG functional pathways in sugarcane’s PICRUSt predicted metagenome were showed by using LEfSe, lineages with LDA values higher than 2.5 were displayed.

**Figure 7 ijms-23-10303-f007:**
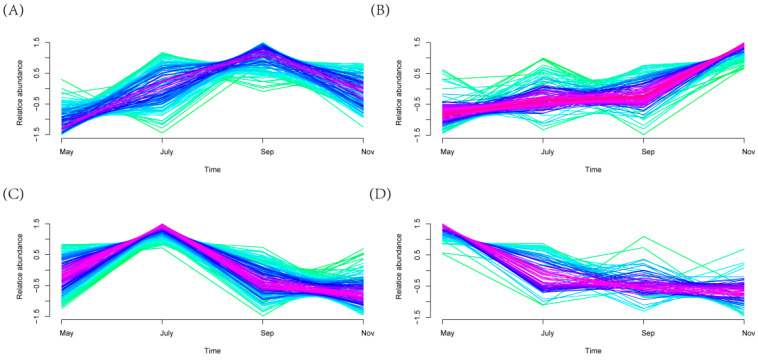
Clustering results of Cluster (**A**) 1, (**B**) 2, (**C**) 3 and (**D**) 4. Red shades indicate high membership values and green shades low membership values of bacterial.

**Figure 8 ijms-23-10303-f008:**
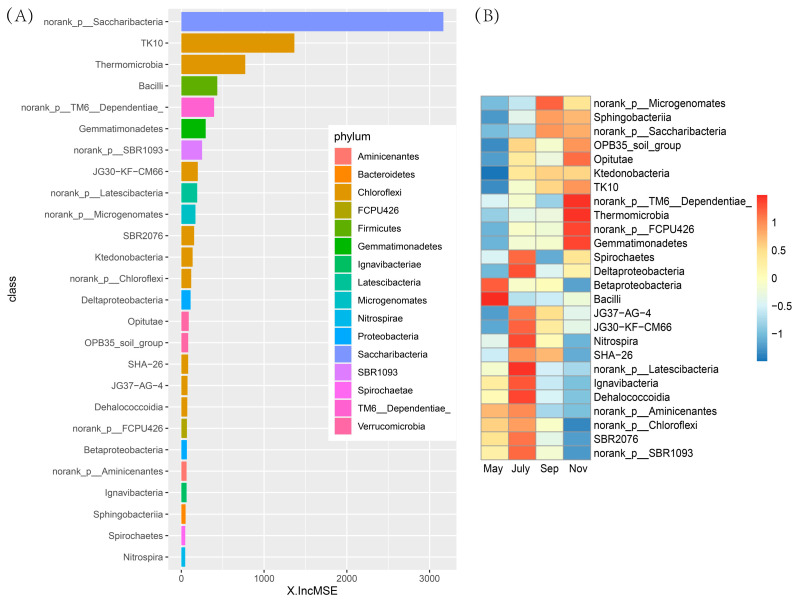
Bacterial taxonomic biomarkers of sugarcane in different stages in fields. (**A**) The top 26 biomarker bacterial classes were identified by applying Random Forests regression of their relative abundances. Biomarker taxa are ranked in descending order of importance to the accuracy of the model, the abscissa represents the weight calculation of each bacterial category indicator. (**B**) Heatmap showing the relative abundances of the top 26 biomarker bacterial classes in the rhizosphere soil of sugarcane.

**Table 1 ijms-23-10303-t001:** Anosim analysis between sugarcane growth stages.

	May		July		Sep	
	*R* ^2^	*p*	*R* ^2^	*p*	*R* ^2^	*p*
July	0.967	0.001 **				
Sep	0.972	0.001 **	0.382	0.001 **		
Nov	0.952	0.001 **	0.618	0.001 **	0.373	0.001 **

Note: Analysis of similarity was calculated between all treatments based on OTUs tables Bray–Curtis distance matrices. Each pairwise comparison of two groups was performed using 999 permutations. R^2^ values > 0.75 are generally interpreted as clearly separated, R^2^ > 0.5 as separated and R^2^ < 0.25 as groups hardly separated. ** *p* < 0.01.

## Data Availability

The raw sequencing data has been deposited in the public database NCBI, registration number: PRJNA721464.

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
