# Peer review of "Sugarcane Rhizosphere Bacteria Community Migration Correlates with Growth Stages and Soil Nutrient"

_ijms, 2022, doi:10.3390/ijms231810303_

Round 1
Reviewer 1 Report
The manuscript by Yuan et al., reports changes in the bacteria community during different developmental stages of the sugarcane. The authors found changes in the bacteria community according to a particular developmental stage. Additionally, the authors observed that the composition of the bacteria community is modified by the presence of certain nutrients as well as with pH. This is an interesting study that prodives evidence of how the bacteria community is shaped based in the presence or certain plant species. There are some aspects that must be addressed:
1) Line 99: The authors mention that TB was significant improved during the warmer months. However, I can see a significant decrease. Please clarify
2) Line 100, what is TM means?
3) Do the authors know the bacteria community before planting? It is important to know the original or before planting bacteria community. This information will help to understand how the community changes during the sugarcane development
Author Response
Reviewer 1
The manuscript by Yuan et al., reports changes in the bacteria community during different developmental stages of the sugarcane. The authors found changes in the bacteria community according to a particular developmental stage. Additionally, the authors observed that the composition of the bacteria community is modified by the presence of certain nutrients as well as with pH. This is an interesting study that prodives evidence of how the bacteria community is shaped based in the presence or certain plant species. There are some aspects that must be addressed:
1) Line 99: The authors mention that TB was significant improved during the warmer months. However, I can see a significant decrease. Please clarify
Response: Thank you very much for pointing out the error in our soil results section, the TN described in the original text should be TP, and the TP and OM indicators of the rhizosphere soil improved significantly between the early growth period and the elongation period, which corresponds to Figure 1B, C and E. We apologize for the confusion caused by our incorrect writing, and we have corrected such errors in the original text.
2) Line 100, what is TM means?
Response: Thank you very much for pointing out our mistake, the TM in the original text should be Tem, which stands for soil temperature environmental factor. We are very sorry for the confusion caused by our wrong writing, and have corrected it in the original text and marked it in yellow.
3) Do the authors know the bacteria community before planting? It is important to know the original or before planting bacteria community. This information will help to understand how the community changes during the sugarcane development.
Response: Thank you very much for your question. As you said, the structure of the soil microbial community before the start of the experiment is very important. Our experiment was conducted after the sugarcane had been grown for a certain period of time and its rhizosphere community had stabilized, and the purpose of the study was to discuss the changes in the rhizosphere flora during the sugarcane production cycle, but we believe that the questions you raised are very important to include in our subsequent study.
Reviewer 2 Report
This work investigated seasonal variations in rhizosphere bacteria community of sugarcane via high-throughput sequencing, providing knowledge on the the interaction of rhizosphere microorganisms with plants. The Some issues are to be further addressed.
1. The introduction section is too long. Please separate it into several paragraphs. The same thing to the Discussion section.
2. Line 417: 16S rRNA.
3. The novelty of this work should be further explained, as extensive similar work has been done.
4. The samples are taken within one year, do the authors consider the viariation among different years?
Author Response
Reviewer 2
This work investigated seasonal variations in rhizosphere bacteria community of sugarcane via high-throughput sequencing, providing knowledge on the the interaction of rhizosphere microorganisms with plants. The Some issues are to be further addressed.
- The introduction section is too long. Please separate it into several paragraphs. The same thing to the Discussion section.
Response: Dear Reviewer, thank you very much for your suggestion. We have divided the introduction into four main sections and the discussion into five main viewpoint sections based on the content of the study.
- Line 417: 16S rRNA.
Response: Thank you very much for pointing out our mistakes, we have changed to check for similar errors in the article.
- The novelty of this work should be further explained, as extensive similar work has been done.
Response: We appreciate your valuable suggestions, and we have added a more detailed description of our innovations at the end of the preface, including but not limited to noting how we differ from other similar efforts. At the same time, through this research work, we hope to provide a more comprehensive and extensive basis for modeling plant root-microbe interactions.
- The samples are taken within one year, do the authors consider the viariation among different years?
Response: Thank you very much for your question about the time of sample collection in the study. For the effect of different years on sugarcane rhizosphere bacteria, we sampled the samples after the sugarcane cultivation reached a certain stable period of time, and as for the effect between years, we also collected the soil from the same location of sugarcane cultivation for 10, 15 and 30 years in the follow-up experiments for more detailed study of the effect of years on the sugarcane rhizosphere micro-environment.
Reviewer 3 Report
This study brings a valuable information on the issue. With a very concise and attractive introduction and a very didactic and technical graphic presentation, the authors capture readers. For improving the text, I kindly share with you my suggestions:
Line 81 – provide the year of publication of Zhang et al.
Section 2 – considering readers around the globe, please provide the information of season, as follows: July (summer), November (fall). Authors should note that sugarcane is a widespread crop and in the southern hemisphere July is winter and November (spring). It is also good to understand results, for example, more OM during the warmer months as well as more microbial diversity.
Section 2.5 – please confirm the use of names not italicized for family taxa (lines 228-230).
Line 296 – please provide the year of publication of Yuan et alli study.
Lines 303-304 – please check the suggestion as same used in section 2.5 (for the phyla)
Line 326 – please follow the same observation used for line 296.
Line 328 – same as line 326.
Line 346 – please see the observation for section 2.5
Line 253 – Deltaproteobacteria is not italicized
Lina 364-365 – same as section 2.5
Line 375 – same as section 2.5
Line 395 – same as line 326
Line 475-476 – same as section 2.5
In my opinion, a short discussion must be done in term of consideration the use of agrochemicals in the land. The information of presence of agrochemicals should be provided when soil characteristics are listed. Was it an organic crop or the conditions studied have considered the presence of chemicals. What do the agrochemicals may interfere with microbial diversity?
The study also brings important ecological information and I suggest discuss how in tropical zones the microbial dynamic may differ than crops in the north or south temperate zones. You may compare with previous studies (if the behaviour is equal or different than diversity you found in your study)
Please strongly check the use of italicized names in all manuscript’s text. They are used only for domain, genera and species.
Author Response
Reviewer 3
This study brings a valuable information on the issue. With a very concise and attractive introduction and a very didactic and technical graphic presentation, the authors capture readers. For improving the text, I kindly share with you my suggestions:
Line 81 – provide the year of publication of Zhang et al.
Response: Thank you very much for your questions regarding the writing specifications of our paper. We have made all the corrections for the similar problem of not adding the year to the cited literature in the article through careful checking, and thank you again for your very constructive comments.
Section 2 – considering readers around the globe, please provide the information of season, as follows: July (summer), November (fall). Authors should note that sugarcane is a widespread crop and in the southern hemisphere July is winter and November (spring). It is also good to understand results, for example, more OM during the warmer months as well as more microbial diversity.
Response: Thank you very much for your suggestion, which is very important for a global audience, and I am sorry for the shortcomings of our article writing, we have added a description of the seasons in the sampling section of the material methods, and thank you again for your very constructive suggestions.
Section 2.5 – please confirm the use of names not italicized for family taxa (lines 228-230).
Response: Thank you very much for pointing out our errors. We have carefully re-examined the use of italics throughout the text and have standardized the names of bacteria except at the genus and species levels.
Line 296 – please provide the year of publication of Yuan et alli study.
Response: Thank you very much for your question, and we apologize for the error regarding the year information not being added to the cited literature. Similar errors have been carefully checked and verified, and have been corrected in the text and highlighted in yellow.
Lines 303-304 – please check the suggestion as same used in section 2.5 (for the phyla)
Response: Thank you very much for pointing out the errors in our articles, such errors have been carefully checked and corrected in our articles.
Line 326 – please follow the same observation used for line 296.
Response: Thank you very much for pointing out the errors in our articles, such errors have been carefully checked and corrected in our articles.
Line 328 – same as line 326.
Response: We are very sorry for such errors, we have checked and corrected them in the whole text
Line 346 – please see the observation for section 2.5
Response: We are very sorry for such errors, we have checked and corrected them in the whole text
Line 253 – Deltaproteobacteria is not italicized
Response: Thank you very much for finding and pointing out our mistake, we have corrected the incorrect use of italics in the original text.
Lina 364-365 – same as section 2.5
Response: We are very sorry for such errors, we have checked and corrected them in the whole text
Line 375 – same as section 2.5
Response: We are very sorry for such errors, we have checked and corrected them in the whole text
Line 395 – same as line 326
Response: We are very sorry for such errors, we have checked and corrected them in the whole text
Line 475-476 – same as section 2.5
Response: We are very sorry for such errors, we have checked and corrected them in the whole text
In my opinion, a short discussion must be done in term of consideration the use of agrochemicals in the land. The information of presence of agrochemicals should be provided when soil characteristics are listed. Was it an organic crop or the conditions studied have considered the presence of chemicals. What do the agrochemicals may interfere with microbial diversity?
Response: Thank you very much for your suggestion about the effect of pesticide use and residues on sugarcane rhizosphere soil microorganisms which we do not know yet, but we still added a short discussion about soil pesticide residues in the discussion section based on your suggestion, which is indeed a very important issue, and we tried to avoid the use of pesticides during sugarcane planting and growth management in this study in order to eliminate the effect of this aspect on the experimental results. What agrochemical samples affect the diversity of the microbial community will be fully considered in our subsequent studies.
The study also brings important ecological information and I suggest discuss how in tropical zones the microbial dynamic may differ than crops in the north or south temperate zones. You may compare with previous studies (if the behaviour is equal or different than diversity you found in your study)
Response: Thank you very much for your suggestion, which is important for improving the scientific integrity and consistency of our study, and we have added a short discussion about it in the Discussion section based on your suggestion.
Please strongly check the use of italicized names in all manuscript’s text. They are used only for domain, genera and species.
Response: We apologize for the incorrect use of italics in the article, and we have corrected such problems in the original article after careful scrutiny. Thank you very much for your constructive questions to the article.
Round 2
Reviewer 1 Report
The authors have addressed my comments